# Discovery of small molecule agonists of the Relaxin Family Peptide Receptor 2

Maria Esteban-Lopez[1], Kenneth J. Wilson[2], Courtney Myhr[1], Elena M. Kaftanovskaya[1], Mark J. Henderson[2], Noel T. Southall [2], Xin Xu[2], Amy Wang[2], Xin Hu[2], Elena Barnaeva[2], Wenjuan Ye[2], Emmett R. George[2], John T. Sherrill [3], Marc Ferrer[2], Roy Morello[4], Irina U. Agoulnik [1,5], Juan J. Marugan [2✉] & Alexander I. Agoulnik [1,5✉]

The relaxin/insulin-like family peptide receptor 2 (RXFP2) belongs to the family of class A G-protein coupled receptors (GPCRs) and it is the only known target for the insulin-like factor 3 peptide (INSL3). The importance of this ligand-receptor pair in the development of the gubernacular ligament during the transabdominal phase of testicular descent is well established. More recently, RXFP2 has been implicated in maintaining healthy bone formation. In this report, we describe the discovery of a small molecule series of RXFP2 agonists. These compounds are highly potent, efficacious, and selective RXFP2 allosteric agonists that induce gubernacular invagination in mouse embryos, increase mineralization activity in human osteoblasts in vitro, and improve bone trabecular parameters in adult mice. The described RXFP2 agonists are orally bioavailable and display favorable pharmacokinetic properties, which allow for future evaluation of the therapeutic benefits of modulating RXFP2 activation in disease models.

[1] Department of Human and Molecular Genetics, Herbert Wertheim College of Medicine, Florida International University, Miami, FL, USA. [2] Early Translation Branch, National Center for Advancing Translational Sciences, National Institutes of Health, Bethesda, MD, USA. [3] Department of Orthopaedic Surgery, University of Arkansas for Medical Sciences, Little Rock, AR, USA. [4] Department of Physiology & Cell Biology, University of Arkansas for Medical Sciences, Little Rock, AR, USA. [5] Biomolecular Sciences Institute, Florida International University, Miami, FL, USA. ✉email: maruganj@nih.gov; alexander.agoulnik@fiu.edu

NSL3 is a peptide hormone of the relaxin/insulin-like family produced in the testicular Leydig cells and ovarian theca cells[1,2]. The biological function of the INSL3/RXFP2 receptor pair was first discovered in the early 2000s when *Insl3*−/− or *Rxfp2*−/− male mice were shown to develop cryptorchidism, or undescended testes, characterized by the failure of the male gonad to descend into the scrotal position during embryonic development. In the mutant mice, testes remain in the intrabdominal cavity close to the kidneys. The *Insl3*−/− and *Rxfp2*−/− mice presented with an underdeveloped inguinoscrotal ligament, the gubernaculum, demonstrating the importance of the INSL3/RXFP2 ligand-receptor pair in gubernacular differentiation and growth in embryonic development during the initial stage of transabdominal descent of the testes towards their future scrotum position[3–7]. In humans the undescended testis is the most common congenital birth defect, affecting 1–3% of full-term newborn boys[8]. While some of the milder cases of inguinal maldescent are resolved spontaneously, this condition otherwise requires surgical correction of the testis position[9]. There is no reliable pharmacological treatment of this condition. Mutation analysis of patients with undescended testes have identified several loss-of-function mutations of *INSL3* and *RXFP2* genes, which further suggests the involvement of this ligand-receptor pair in cryptorchidism[3,10–15].

The high expression of INSL3 is maintained in postnatal and adult testes, raising the question of its role in reproductive and other organs. The serum concentration of INSL3 correlates with normal spermatogenesis and decreases with age in men[16]. Women have much lower serum levels of INSL3, which is produced in adult ovarian theca cells and fades after menopause[17]. In recent years, INSL3 and RXFP2 have also been implicated in maintaining bone homeostasis. Mutations of the RXFP2 receptor in human patients with cryptorchidism have been linked to the development of osteopenia or osteoporosis. A similar phenotype of decreased bone mass was shown in *Rxfp2*−/− mice[18]. RXFP2 expression has been found in osteoblasts but not in osteoclasts[19]. Treatment of primary human osteoblasts with INSL3 has been shown to increase the gene expression of important bone formation markers such as *ALP* and *COL1A1*, as well as improve mineralization of the osteoblast extracellular matrix[20]. Collectively, these data suggest that activation of the RXFP2 receptor has an anabolic effect in bone and could be a pharmacological target for increasing bone formation. Notably, there are limited treatment options for bone diseases such as osteoporosis. Osteoporosis treatments work by either inhibiting osteoclast bone resorption or by promoting osteoblast bone formation, both of which result in increased bone mass. However, most of drugs are not orally bioavailable and rare but severe side effects have been documented after chronic use[21–26]. While RXFP2 is a potential target to increase bone formation, the use of INSL3 peptide as a therapeutic agent is not practical because the recombinant hormone has a short half-life in vivo and lacks oral bioavailability.

To address these limitations, we performed a high-throughput screening campaign to identify RXFP2 agonists using the NCATS/NIH small molecule compound library. The hit compounds identified in the screening were further optimized to develop a set of molecules with improved potency, selectivity, oral bioavailability, osteoblast mineralization activity in vitro, and biological activity in vivo. The restricted expression of RXFP2 combined with the high specificity of the small molecule agonists provides an opportunity for systemic delivery of these compounds without serious side effects. Hence, activation of RXFP2 with small molecule agonists might further elucidate the physiological role of this receptor signaling in vivo and represents a novel pharmacological strategy for treating reproductive diseases and diseases associated with bone loss.

## Results

### Identification and optimization of small molecule agonists of human RXFP2.

A quantitative high-throughput screening (HTS) of more than 80 K compounds of the NCATS/NIH diversity collection was performed at four concentrations in HEK293T cells stably expressing human RXFP2 to identify candidate agonists. A homogenous time resolved fluorescence (HTRF) assay was used to directly detect compound-induced cAMP accumulation, with forskolin or human recombinant INSL3 serving as positive controls. Approximately 150 active compounds ($EC_{50} < 10\,\mu M$, efficacy > 40% of 10 nM INSL3, robust dose-dependent agonism, and no HTRF reagent artifacts) were re-synthesized and re-tested in the same assay. Selected compounds with reasonable activity in dose response were also tested in a confirmatory orthogonal screening using a cAMP response element (CRE) driven luciferase reporter assay in HEK-CRE-Luc-RXFP2 cells, and in a counter-screen with cells transfected with human relaxin/insulin-like family peptide receptor 1 (RXFP1). Finally, we performed an extensive structure activity relationship campaign (SAR) to increase both the agonist efficacy and potency of the lead series. Optimized molecules displayed high potency and efficacy in the cAMP assay. The cAMP responses for several lead compounds from the HTS are shown in Fig. 1. All compounds had $EC_{50}$ below 1 µM and are ranked by mean $E_{max} \pm SEM$ as follows: 6641 (106.99 ± 8.77), 4337 (78.53 ± 18.81), 4340 (53.14 ± 4.88) and 1715 (41.32 ± 3.58). A confirmatory CRE-luciferase cAMP screen was used to eliminate any potential false positives due to assay detection artifacts (Supplementary Fig. 1a).

Our prior work on RXFP1 revealed species-specific agonism for identified molecules[27]. Therefore, we examined the capacity of the identified human RXFP2 agonists to activate murine RXFP2. A CRE-luciferase reporter assay in cells expressing mouse RXFP2 demonstrated that the compounds 6641, 4337, and 4340 also activate the mouse RXFP2 receptor (Supplementary Fig. 1b) suggesting that the receptor binding sites for the small molecule agonists are conserved between species. However, in these CRE-luciferase reporter assays, only compound 6641 behaves as a full agonist of both human (Supplementary Fig. 1a) and mouse RXFP2 (Supplementary Fig. 1b). The observed difference between CRE-luciferase reporter assays with human and mouse RXFP2 might be due to a lower expression of the receptor in transiently transfected mouse RXFP2 cells compared to the stably transfected human RXFP2 cell line.

### Small molecule agonists are selective for human RXFP2.

The RXFP1 receptor has the highest amino acid sequence identity with RXFP2, which is around 60%[28,29]. To demonstrate the specificity and selectivity of the identified small molecule RXFP2 agonists, a HEK293T cell line stably transfected with human or mouse RXFP1 was used. The results from the HTRF cAMP assay using the RXFP1-expressing cell lines demonstrated that compounds 6641, 4337, and 4340 do not activate either human or mouse RXFP1 receptors (Supplementary Fig. 2). The related RXFP3 and RXFP4 subgroup only share around 10% sequence identity with RXFP2, and their activity is mediated by $G\alpha_i$ protein coupling[30]. A GlowSensor cAMP assay in RXFP3 or RXFP4 transiently transfected HEK293T cells demonstrated that compound 6641 does not activate these receptors (Supplementary Fig. 3). The class A glycoprotein hormone receptors LHCGR, FSHR, and TSHR share around 30% amino acid sequence identity with RXFP2, and they all activate $G\alpha_s$ signaling[30]. However, treatment of cells transfected with these receptors with 6641 did not result in an increase of cAMP level analyzed using a Hit Hunter cAMP assay (Supplementary Fig. 4). The PTHR1 receptor is a member of the class B GPCRs, also signaling by coupling to

**a**

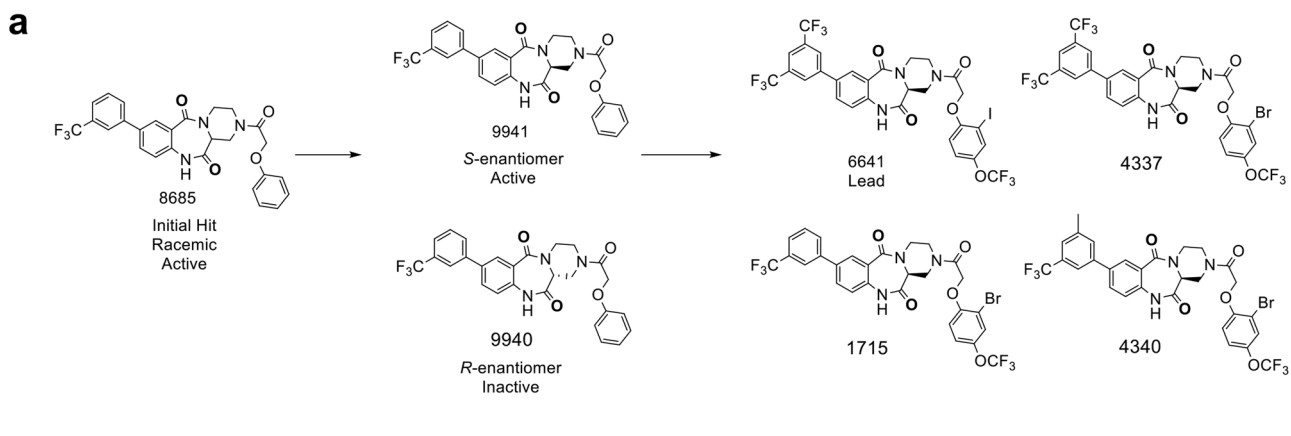

**b**

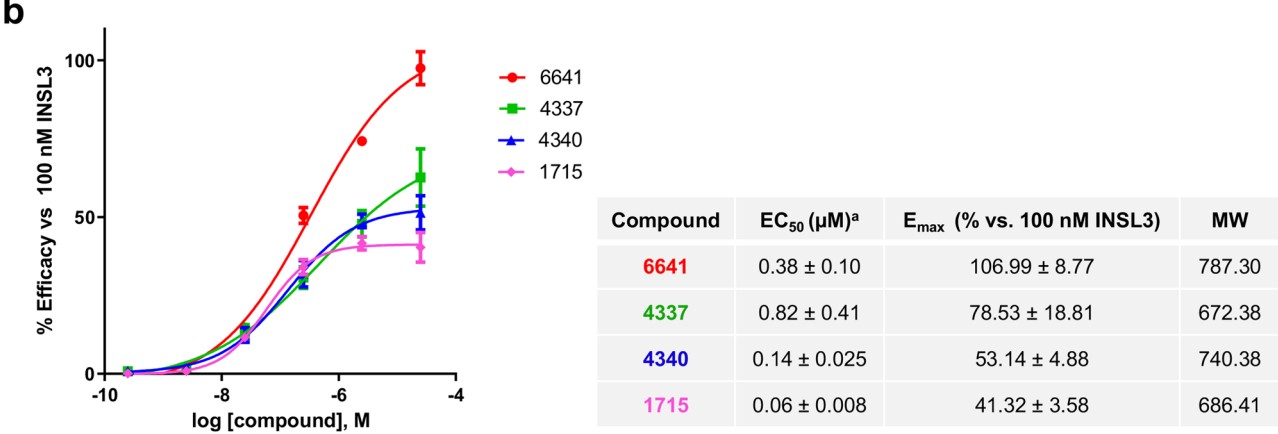

| Compound | EC$_{50}$ (µM)$^a$ | E$_{max}$ (% vs. 100 nM INSL3) | MW |
|----------|---------------------|-------------------------------|--------|
| 6641 | 0.38 ± 0.10 | 106.99 ± 8.77 | 787.30 |
| 4337 | 0.82 ± 0.41 | 78.53 ± 18.81 | 672.38 |
| 4340 | 0.14 ± 0.025 | 53.14 ± 4.88 | 740.38 |
| 1715 | 0.06 ± 0.008 | 41.32 ± 3.58 | 686.41 |

**Fig. 1 RXFP2 agonist screening and SAR study. a** Schematic representation of SAR study showing the initial hit and active enantiomer configuration that led to the final 4 compounds, including the lead compound 6641. **b** Ranking of the RXFP2 agonist activity by HTRF cAMP assay in HEK-RXFP2 cells. The cAMP response induced by the compounds was normalized to the cAMP response induced by 100 nM INSL3 as 100% efficacy and the cAMP response induced by DMSO vehicle as 0% efficacy. E$_{max}$ and EC$_{50}$ were calculated using four parameter nonlinear fit dose-response stimulation curves. Results are expressed as mean ± SEM of 3 independent experiments. $^a$The EC$_{50}$ is expressed in micromolar and is the concentration necessary to reach 50% of maximum cAMP signal. Molecular weight (MW).

Gα$_s$, and although it is not highly homologous with RXFP2, it was also tested to demonstrate the specificity of compound 6641 due to the importance of PTHR1 signaling during bone formation (Supplementary Fig. 4).

β-arrestin is an important GPCR adaptor protein that mediates several receptor processes including signaling, desensitization, and trafficking[31]. The PRESTO-Tango GPCRome assay was further used to test specificity and selectivity of lead RXFP2 agonists 6641 and 4337. GPCR-mediated β-arrestin recruitment for a panel of 320 human GPCRs was studied in transiently transfected HTLA cells treated with 10 µM compound. The results for all 320 GPCRs tested are summarized in Supplementary Data 1. Compound 6641 was found to induce β-arrestin recruitment above the basal level of detection (3-fold RLU) for 3 GPCRs (mean ± SEM): ADORA1 (3.92 ± 0.15), CCKAR (3.42 ± 0.15) and FPR1 (3.11 ± 0.29). Compound 4337 activated β-arrestin recruitment in 7 GPCRs: ADORA1 (5.24 ± 0.31), FPR1 (4.89 ± 0.12), GPR20 (4.22 ± 0.91), CCKAR (3.57 ± 0.17), CXCR7 (3.44 ± 0.48), P2RY1 (3.07 ± 0.14) and MRGPRF (3.01 ± 0.19). A follow-up test of ADORA1, FPR1, GPR20, and CCKAR using multiple compound concentrations revealed that only the adenosine A1 receptor (ADORA1) responded to compound 6641 and 4337 at concentrations above 1 µM, but with much lower efficacy than the ADORA1 agonist 5′-(N-Ethylcarboxa-mido)adenosine (NECA) (Supplementary Fig. 5). ADORA1 activation by an agonist results in Gα$_i$ protein binding, an inhibition of adenylate cyclase and, therefore, a decrease in the cAMP concentration. Analysis of cAMP inhibition after treatment of ADORA1-HEK293T transfected cells with both compounds failed to detect any significant changes (Supplementary Fig. 6). Thus, while compounds appear to bind to ADORA1, they do not trigger receptor activation. Of note, RXFP2-mediated β-arrestin recruitment induced by compounds 6641 and 4437 was not detected by the PRESTO-Tango assay. This finding agrees with previous data showing a lack of RXFP2-mediated β-arrestin recruitment after treatment with INSL3[32]. All together, these results identified compound 6641 as a highly selective RXFP2 agonist.

**Compound 6641 is an allosteric agonist of human RXFP2.** Small molecule agonists of GPCRs often bind to allosteric receptor sites to induce downstream signaling. In many cases, this may result in biased cellular signaling with more specific therapeutic activity and fewer side effects[33,34]. Here, we analyzed the mechanisms of receptor activation by compound 6641 in comparison to the natural ligand INSL3 to assess if it is indeed an allosteric agonist of RXFP2. The current INSL3/RXFP2 binding model shows that the B-chain of the INSL3 heterodimer binds with high affinity to the extracellular leucine-rich repeat (LRR) domain of RXFP2[35–37]. Conversely, the INSL3 B-chain homodimer is an RXFP2 antagonist which binds to the LRR domain of RXFP2 without inducing a cAMP response[38,39]. Co-treatment

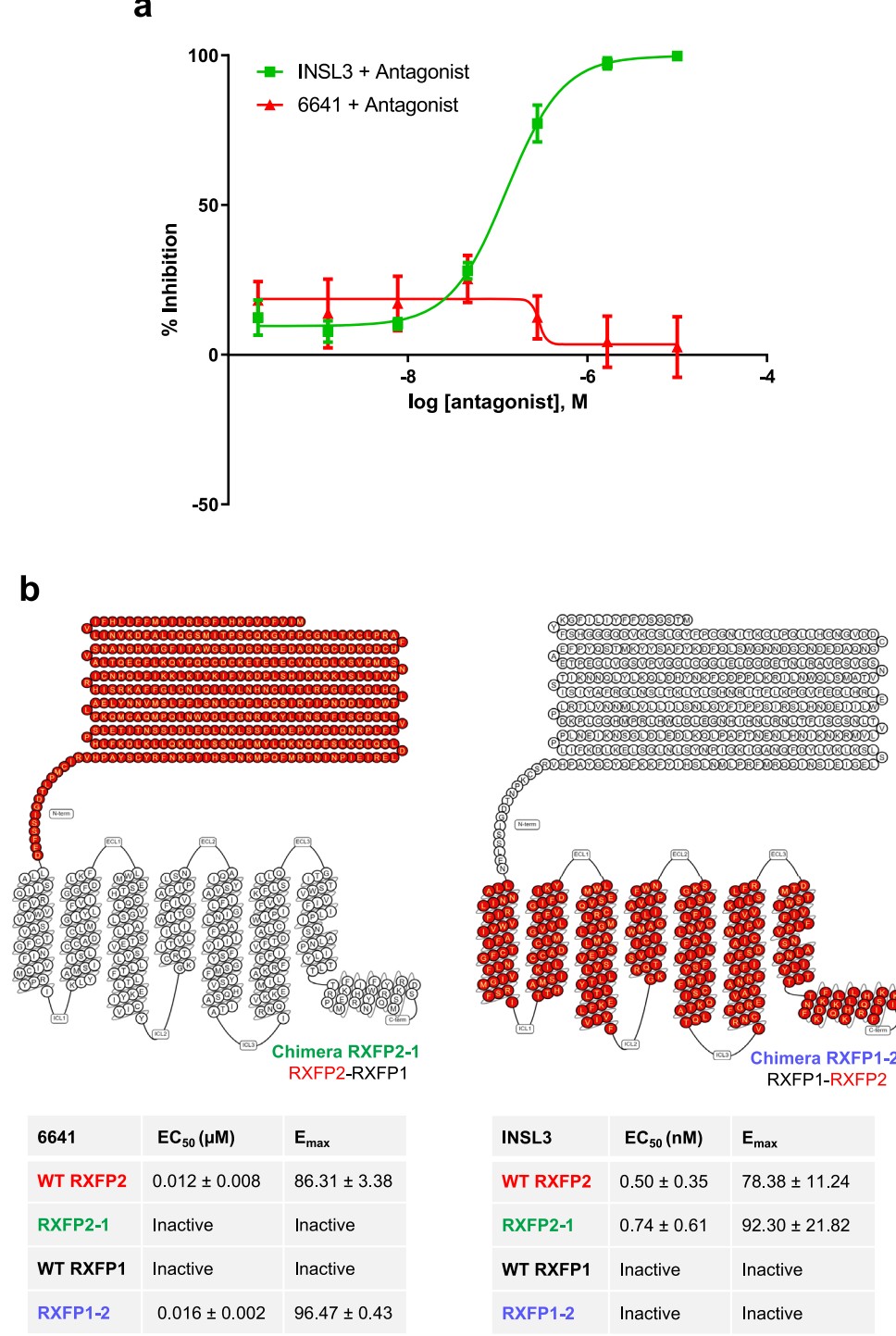

**Fig. 2 6641 is an allosteric agonist of RXFP2. a** HEK-RXFP2 cells co-treated with serial dilutions of INSL3 B dimer antagonist and 30 nM INSL3 or 0.28 μM 6641. HTRF cAMP dose-response inhibition curves were normalized to activity without antagonist co-treatment as 0% inhibition. **b** HTRF cAMP assay in HEK293T cells transiently transfected with WT or chimeric receptors. Chimera RXFP2-1 contains the extracellular domain of RXFP2 and the transmembrane domain of RXFP1, while chimera RXFP1-2 contains the extracellular domain of RXFP1 and the transmembrane domain of RXFP2. For each receptor, cAMP response induced by INSL3 and 6641 was normalized to FSK (2 μM) as 100% efficacy and DMSO vehicle as 0% efficacy. $E_{max}$ and $EC_{50}$ were calculated using four parameter nonlinear fit dose-response curves. Results are expressed as mean ± SEM of 3 independent experiments. Diagrams of the chimeric receptors were produced using GPCRdb.org[58, 59].

with RXFP2 antagonist caused a dose-dependent inhibition of INSL3/RXFP2-mediated cAMP response but had no inhibitory effect on the 6641/RXFP2-induced cAMP response (Fig. 2a).

As shown above, the identified small molecule compounds are selective agonists of the RXFP2 receptor and do not activate

RXFP1. Complementary RXFP2-1 and RXFP1-2 chimeric receptors were used to further investigate the involvement of the RXFP2 extracellular and transmembrane domains on 6641 activity. RXFP2-1 has the extracellular domain of RXFP2 and transmembrane domain of RXFP1, and RXFP1-2 is its

complement[40]. In the HTRF cAMP assay, compound 6641 did not stimulate cAMP production in cells transfected with the chimera RXFP2-1 but induced a cAMP response in the chimera RXFP1-2 ($EC_{50} = 0.016 \pm 0.002 \mu M$, Emax = 96.47 ± 0.43). On the contrary, INSL3 induced a cAMP response in the chimera RXFP2-1 ($EC_{50} = 0.74 \pm 0.61$, $E_{max} = 92.30 \pm 21.82$) but not RXFP1-2 (Fig. 2b). The chimeric receptors showed similar surface expression to each other and the wild-type RXFP1 and RXFP2 receptors (Supplementary Fig. 7). Thus, the results indicate that the small molecule 6641 is an allosteric agonist of RXFP2 that requires the RXFP2 transmembrane domain for activity.

We have further analyzed the effect of simultaneous activation of human RXFP2 with 6641 and INSL3 to evaluate possible allosteric modulation of INSL3 cAMP signaling by the small molecule (Supplementary Fig. 8). An HTRF cAMP assay showed that co-stimulation by the two ligands had a simple additive effect on cAMP production. Thus, compound 6641 is not acting as either a positive or negative allosteric modulator of INSL3 but behaves as a full RXFP2 allosteric agonist.

**RXFP2 agonists specifically activate the mouse RXFP2 receptor in vivo**. The INSL3 surge that drives gubernaculum development during the transabdominal stage of testicular descent in male mice occurs between embryonic day E14.5 and 17.5[5,6,41]. In females, there is no detectable INSL3 expression and therefore the gubernaculum remains underdeveloped as a vestigial structure. It was previously shown that transgenic overexpression of INSL3 in female embryos resulted in partial gubernacular development, descent of the ovaries into a low intraabdominal position and development of an inguinal hernia[42]. Therefore, we hypothesized that the small molecule agonists of RXFP2 would show similar effects in female mice. Compound activity was tested by injecting pregnant females with the RXFP2 agonists starting from day E12.5 to E17.5 of embryonic development followed by histological evaluation of the gubernaculum in developing female embryos. To define the pharmacokinetic properties of our lead compound 6641, an initial study was performed to evaluate the level of exposure in different tissues upon single dose intraperitoneal (IP) injection in female mice at 30 mg/kg. At this dose, compound 6641 provides stable levels in the range of single/double digit μM in plasma, liver, and bone for at least 12 h (Supplementary Fig. 9). At E18.5 the *processus vaginalis* or outpouching of the peritoneum occurs in males at the site of gubernacular caudal connection[43] (Fig. 3a). The female embryos treated with compounds 6641, 4337, and 4340 exhibited a male-like invagination of the gubernaculum into the peritoneum (Fig. 3b). This phenotype was not observed in the untreated and vehicle-treated female controls at this age, suggesting that the small molecule compounds passed through the placental barrier and induced INSL3-like effects in female gubernaculum in vivo. No differences were detected in the position of the ovaries in compound-treated female embryos and controls.

**RXFP2 agonists induce osteoblast mineralization in vitro**. The lead RXFP2 agonists were then tested to determine their ability to induce bone mineralization in vitro. We used primary human osteoblasts since they faithfully reproduce the mineralization process of the bone matrix in vitro[44]. Primary human calvarial osteoblast (HCO) cells, patient-derived osteoblasts isolated from human calvariae, mineralize the secreted matrix in vitro and express endogenous RXFP2.

HCO cells were treated for 14 days and extracellular mineral deposition was quantified by measuring hydroxyapatite, the main mineral component of the bone-like nodules deposited by osteoblasts[45]. The results showed that osteoblast mineralization is significantly increased by compound 6641 at 3 and 5 μM, but no significant changes were observed at 1 μM (Fig. 4a). A significant increase in osteoblast mineralization was also observed with 5 μM of compound 4337 and 4340 treatment, but not at the lower doses. No significant improvements in osteoblast mineralization were found after treatment with compound 1715 (Fig. 4a).

Cell viability studies were undertaken to identify potential cytotoxic effects of the RXFP2 agonists. HCO cells and human aortic vascular smooth muscle cells (HAVSMC) were treated for 24 h. The results indicated that some cytotoxicity in the HCO and HAVSMC cells was observed only at the highest concentration of 25 μM with compounds 6641, 4337, and 4340. At 8.3 μM and below, no toxicity was found in HCO cells and toxicity was below 20% in HAVSMC cells for all three compounds (Fig. 4b, c). The latter result suggests low vascular cytotoxicity of the compounds. Based on these results, compound 6641 was selected as the lead agonist from our series, demonstrating the highest mineralization activity in osteoblasts and no cytotoxicity at therapeutically relevant concentrations.

**INSL3 overexpression induces bone formation in vivo**. To demonstrate that the INSL3/RXFP2 axis can be a pharmacological target to preserve or increase bone mass, as a proof of concept we first used an INSL3 transgenic mouse model overexpressing mouse INSL3. No significant changes in body weight were observed between INSL3 transgenic and wild-type (WT) female mice (Supplementary Fig. 10a). To evaluate the biological activity of circulating INSL3 in transgenic mice overexpressing the peptide, we applied sera isolated from WT and INSL3 transgenic females onto the HEK-RXFP2 cell assay. As expected, peripheral blood collected from transgenic females exhibited significantly higher activation of the RXFP2 expressing cells (Supplementary Fig. 10b). A micro-CT analysis of the trabecular bone in the L4 vertebral body of 22-week-old females revealed a significant increase in the bone volume per tissue volume (BV/TV) of INSL3 transgenic females compared to WT mice (Supplementary Fig. 11a). INSL3 transgenic mice also showed a small increase in trabecular number (Tb.N), trabecular thickness (Tb.Th) and a decrease in trabecular separation (Tb.Sp), although these differences were not statistically significant. Interestingly, connectivity density (Conn.D) and material density (Mat.D) values were significantly higher in INSL3 transgenic mice, which suggest an increase in trabecular connections and mineralization respectively, both indicative of stronger bones (Supplementary Fig. 11a). Quantitative RT-PCR analysis of the tibia showed significantly increased mRNA levels of the osteoblast activity marker *Alp* in INSL3 transgenic mice compared to WT mice, but no differences were found in bone formation markers *Col1a1* and Osteocalcin or osteoblast differentiation marker *Runx2* (Supplementary Fig. 11b).

**Compound 6641 treatment promotes bone formation in vivo**. To better understand the pharmacokinetic properties of compound 6641 and fine tune the adequate dose in our efficacy studies, several pharmacokinetic studies of compound 6641 in female mice were carried out upon single intravenous (IV) dose at 3 mg/kg, single oral (PO) dose at 10 mg/kg and multiple 10 mg/kg PO daily dose administration for 3 consecutive days. On average, the compound exhibited a half-life of between 4–6.5 h depending on the route of administration, with no accumulation at 10 mg/kg. Despite its oral bioavailability around 25–31% (Supplementary Fig. 12a), the molecule displayed good levels of exposure above its cAMP assay $EC_{50}$ in different tissues for extensive

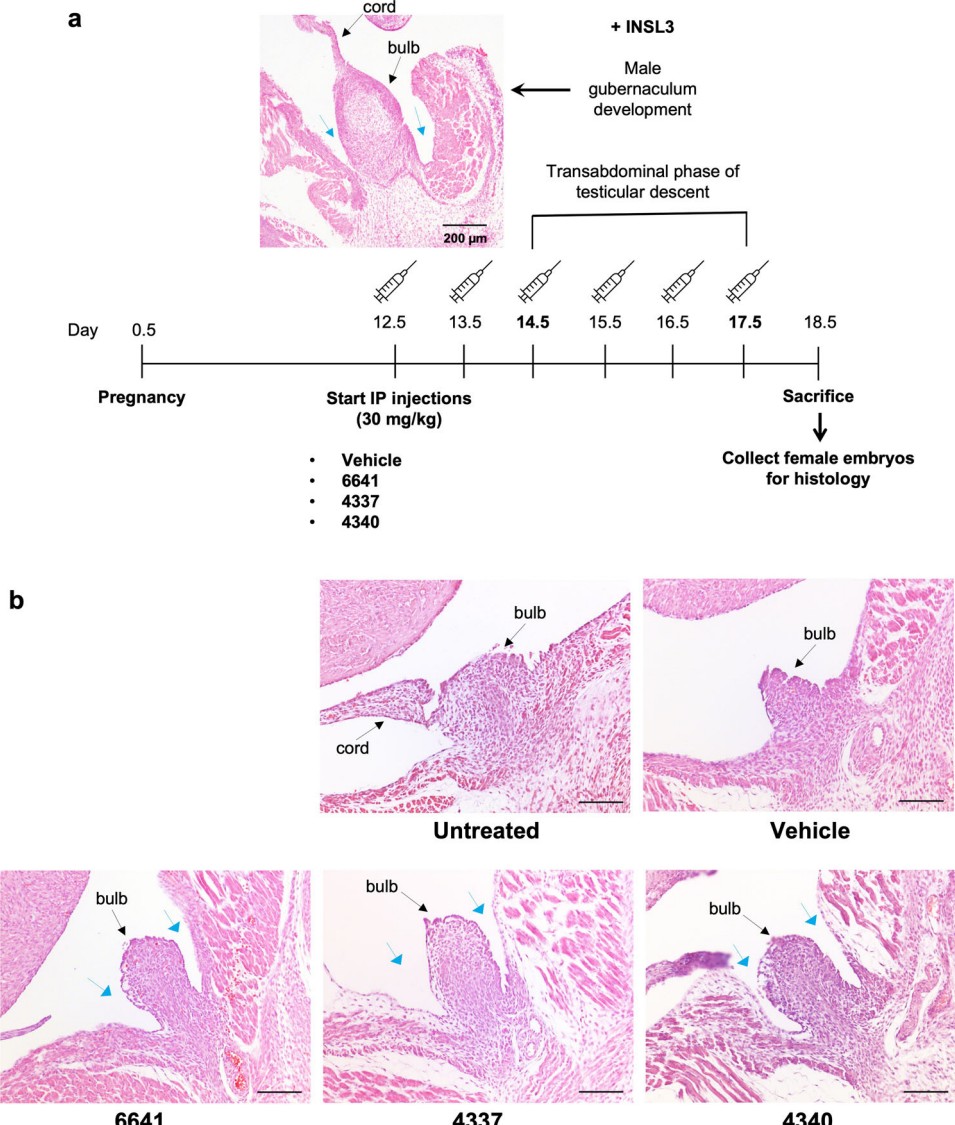

**Fig. 3 RXFP2 agonists are biologically active and selective in vivo. a** Diagram of the treatment protocol and representative histological section of the developed male gubernaculum at embryonic day 18.5 (upper left). **b** Representative histological sections showing a male-like invagination of the female gubernaculum in all compound-treated embryos (blue arrows). Four female embryos were analyzed per group. Scale bar = 100 μm.

periods of time: in bone tissue at a maximum concentration of 1072 ng/g for IV administration and between 170–300 ng/g for PO administration (Supplementary Fig. 12c). Based on this data we selected 10 mg/kg PO administration to carry out our initial efficacy in vivo study, since the dose and route of administration seemed more suitable and clinically relevant for future long-term treatments.

8-week-old WT C57BL/6 J female mice were treated 3 times per week for 8 weeks and the bone anabolic properties of compound 6641 were studied by micro-CT analysis of the trabecular bone in the L3 vertebral body and left femur at the end of the treatment protocol. For the vertebral body, a significant increase in Tb.N and Tb.Th was found in compound-treated mice compared to vehicle-treated mice, while there was a trend toward increased BV/TV and decreased Tb.Sp after compound treatment, respectively (Fig. 5a). No changes were found in Conn.D and Mat.D of the vertebral body between vehicle- and compound-treated mice (Fig. 5a). Micro-CT analysis of the femur showed no changes in any of the trabecular parameters measured after compound treatment (Supplementary

Fig. 13a). A quantitative RT-PCR revealed significantly higher *Rxfp2* gene expression in lumbar spine compared to femur of matching age WT C57BL/6 J female mice (Supplementary Fig. 13b) which could explain the discrepancies observed between micro-CT results in vertebrae and femur. A significant increase in gene expression levels of the osteoblast mineralization marker Osteocalcin and a decrease of the early osteoblast differentiation marker *Runx2* were observed in tibia of compound-treated mice compared to vehicle-treated mice, suggesting a cell population shift from immature osteoblasts to the mature osteoblasts required for bone formation and ossification (Fig. 5b). No significant differences were found in *Alp* and *Col1a1* mRNA levels between groups (Fig. 5b). The body weight of all mice was recorded during treatment, as well as the weight of the liver and kidneys at sacrifice. No changes were observed in body or organ weight between the treatment groups (Supplementary Fig. 14a, b). A histological analysis of the livers and kidneys did not show any changes in cell morphology of compound-treated mice (Supplementary Fig. 14c). No mortality was observed during the study. The results, therefore, suggest that

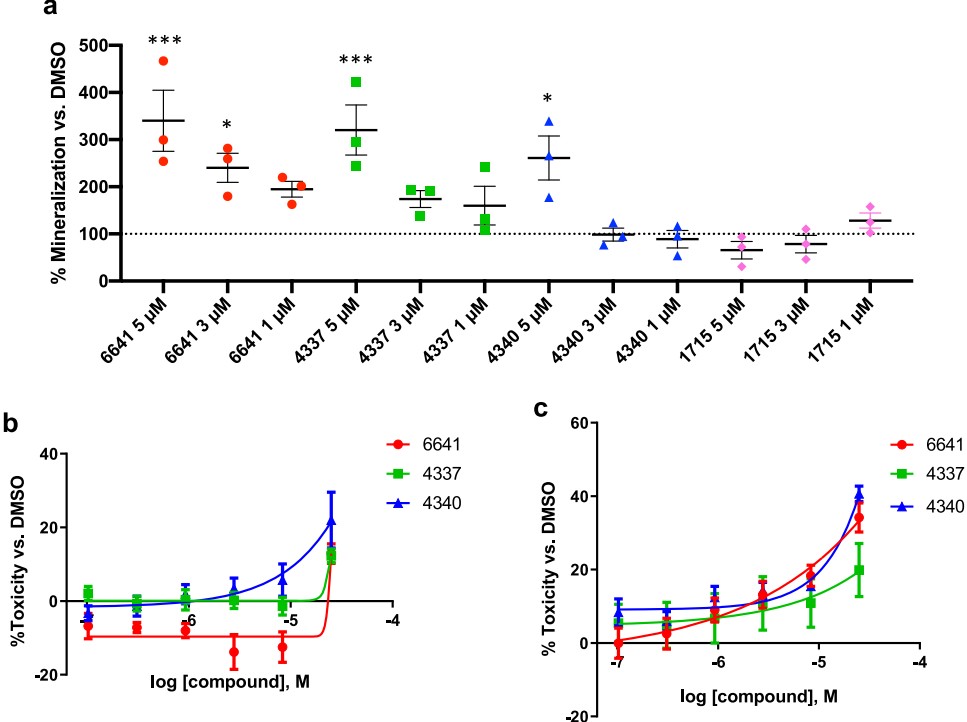

**Fig. 4 RXFP2 agonists induce mineralization of primary human osteoblasts and are non-cytotoxic. a** Mineral hydroxyapatite fluorescent quantification after 14-day treatment in HCO cells. Results were normalized to control DMSO as 100% mineralization. **b** HCO and **c** HAVSMC toxicity after 24-h treatment with RXFP2 agonists, normalized to DMSO treatment as 0% toxicity. Results are expressed as mean ± SEM of 3 independent experiments. *$p < 0.05$, *** $p < 0.001$ vs. DMSO using One-way ANOVA.

there were no major toxicity issues with the PO administration of compound 6641.

## Discussion

In this report we present the discovery of the first-in-class RXFP2 agonists, starting with a HTS screen of more than 80,000 compounds from the Molecular Libraries Small Molecule Repository (MLSMR) at NCATS/NIH against the RXFP2 receptor, and resulting in the identification of a suitable active chemotype for SAR optimization to create agonists with favorable properties for in vivo testing (Fig. 1a). Here, we present 4 of the lead molecules from the SAR study: compounds 6641, 4337, 4340, and 1715. The direct HTRF cAMP assay in HEK-RXFP2 cells showed that the efficacy and activity of compound 6641 ($EC_{50} = 0.38 \, \mu M$, $E_{max} = 107\%$) is greater than the other 3 compounds, with the maximum efficacy of response equivalent to the natural ligand INSL3 (Fig. 1b). A validation screen using HEK-CRE-Luc-RXFP2 cells utilizing a luciferase reporter driven by a CRE-regulated promoter confirmed that all 4 compounds activate RXFP2 receptor signaling. However, due to the accumulation of luciferase, this assay lacks the sensitivity to detect differences in efficacy (Supplementary Fig. 1a). Several counter-screens were implemented to demonstrate the specificity and the absence of off-target effects of the small molecule RXFP2 agonists. Compound 6641 did not activate the highly homologous RXFP1 receptor (Supplementary Fig. 2a) or two other relaxin family protein receptors, RXFP3 and RXFP4 (Supplementary Fig. 3). The two latter receptors utilize $G\alpha_i$ signaling, which inhibits adenylate cyclase and, therefore, induces a decrease in the cAMP concentration. None of these three RXFP receptors were activated by 6641. We also analyzed the effect of 6641 on the cAMP response of related Class A GPCRs, the LRR-containing glycoprotein receptors LHCGR, FSHR, and TSHR, as well as the Class B receptor PTHR1, which is involved in bone formation

(Supplementary Fig. 4). All these receptors signal through $G\alpha_s$ to stimulate cAMP production, but they did not respond to 6641. It should be noted that it was previously shown that activation of RXFP2 results in accumulation of cAMP via $G\alpha_s$, which is then modulated by inhibition of cAMP through the $G\alpha_{oB}$ signaling pathway[46]. Future studies will clarify whether there is biased G protein signaling downstream of RXFP2 after activation with small molecule agonists vs INSL3, as was the case with ML290 and RXFP1[47].

To further investigate possible off-target effects of small molecule agonists of RXFP2 on GPCRs, we performed a PRESTO-Tango assay of 320 GPCRs to detect β-arrestin recruitment to the receptors after treatment with our compounds. Compounds 6641 and 4337 induced β-arrestin recruitment through activation of only one receptor, ADORA1, at concentrations above 1 μM (Supplementary Fig. 5). Importantly, binding of the small molecule agonists to ADORA1 did not result in activation of $G\alpha_i$ signaling (Supplementary Fig. 6). Interestingly, the PRESTO-Tango assay also showed that β-arrestin recruitment was not activated by 6641 or 4337 through the RXFP2 receptor. The results indicate that similar to RXFP1, the RXFP2 receptor could remain activated for long periods of time without undergoing receptor desensitization[32]. This might be beneficial in clinical applications, as it potentially reduces the amount of drug needed to induce a biological response for this target, even in cells with low receptor expression. It should be noted also that a PRESTO-Tango assay is not designed to detect inhibitors of β-arrestin recruitment.

Previous studies of the RXFP1 allosteric agonist ML290 revealed that its interactions with the transmembrane domain of RXFP1 activate the receptor[48,49]. Due to the high structural homology between RXFP1 and RXFP2, it was not surprising to find that the small molecule 6641 is also an allosteric agonist that interacts with the transmembrane domain of RXFP2 (Fig. 2).

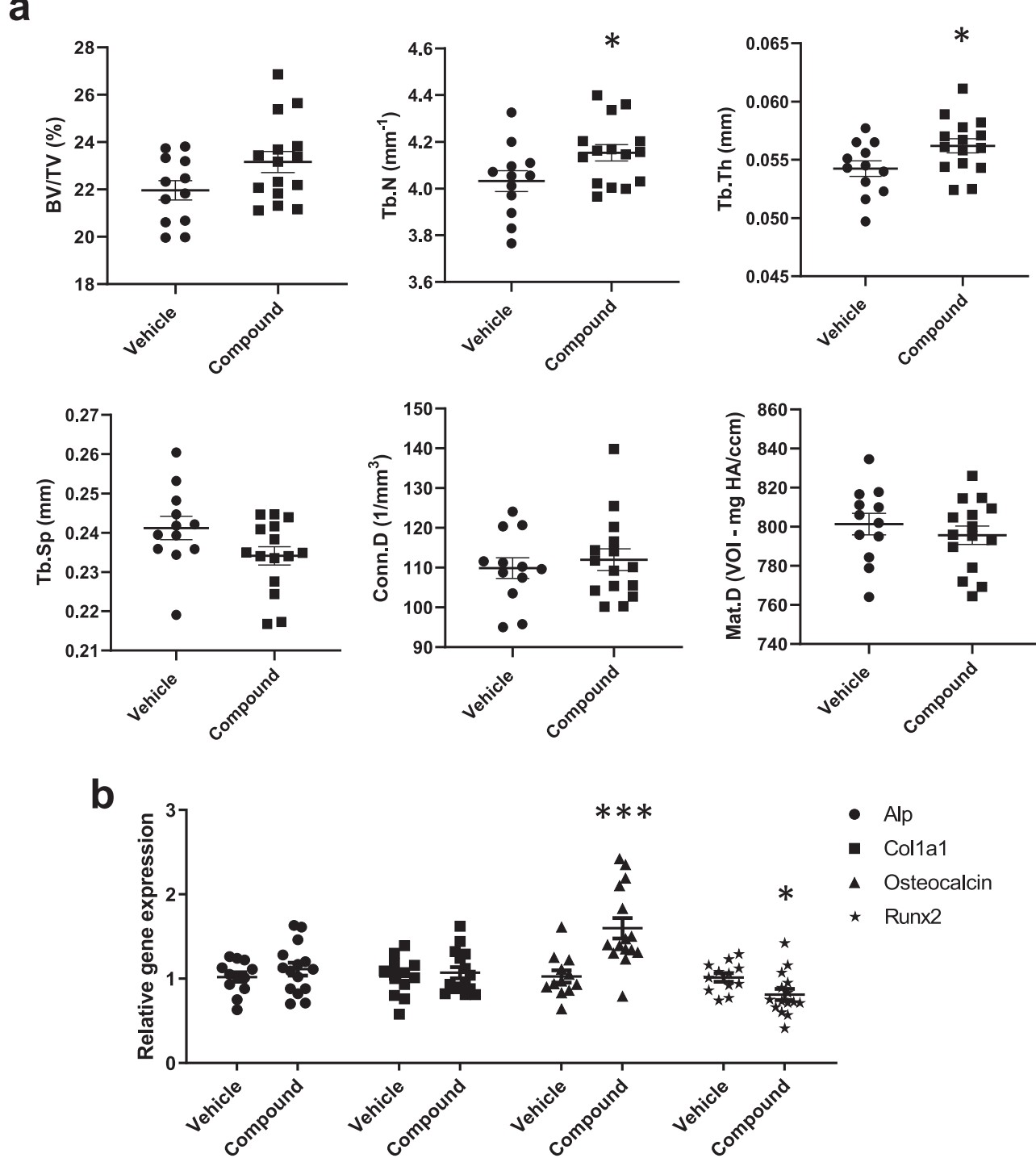

**Fig. 5 Compound 6641 treatment increases bone formation in female mice. a** Trabecular micro-CT parameters in lumbar vertebrae of vehicle- and compound-treated female mice. **b** Gene expression levels of osteoblast markers in tibias from vehicle- and compound-treated female mice measured by quantitative RT-PCR. Twelve and fifteen mice were used in the vehicle and compound groups respectively. qRT-PCR data are normalized to control group and expressed as mean ± SEM. *$p < 0.05$, ***$p < 0.001$ vs. vehicle using Student's $t$-test.

The other question we analyze is whether the small molecule can modulate INSL3 response. Co-stimulation of RXFP2 with both INSL3 and 6641 demonstrated simple additive effects of the two ligands on cAMP production, showing that 6641 is not an RXFP2 modulator. Additionally, in contrast to the species selectivity of RFXP1 observed with compound ML290[48,49], activation of the mouse RXFP2 receptor by agonist compounds was apparent both in vitro (Supplementary Fig. 2b) and in vivo

(Figs. 3, 5). In all these experiments compound 6641 behaved as a full agonist of mouse or human RXFP2 receptor.

Based on the evaluation of cytotoxicity, potency, and cAMP induction efficacy, we selected four compounds for further functional testing in vitro and in vivo. In normal male embryos, it has been described previously that androgen-independent gubernaculum differentiation leads to the *processus vaginalis* or gubernaculum bulb eversion, which is not observed in female

embryos due to the absence of INSL3 expression[50]. Here, we showed that the treatment of pregnant mice with RXFP2 small molecule agonists results in the initiation of the gubernacular eversion in female embryos. As this process is sex- and INSL3/RXFP2-specific, the result strongly suggests that the compounds pass through the placental barrier to the embryos, resulting in RXFP2 activation and gubernacular bulb differentiation (Fig. 3b). The absence of changes in the gubernacular cord and ovary position are most likely due to low compound exposure in comparison to INSL3 in male embryos or in female mice with transgenic overexpression of INSL3[42].

To evaluate the effects of RXFP2 small molecule agonists on bone cells, we first looked at their ability to induce mineralization of primary human osteoblasts expressing RXFP2. We demonstrated that the mineralization activity induced by the compounds 6641, 4337, 4340, and 1715 correlated with their respective cAMP responses observed in HEK-RXFP2 cells (Fig. 4a). This indicates that RXFP2 activation of cAMP signaling is a key cellular pathway that promotes mineralization in osteoblast cells. The importance of cAMP signaling during bone formation has been previously described for the GPCR parathyroid hormone 1 receptor (PTHR1) after activation with the ligand PTH (parathyroid hormone). The PTH/PTHR1 ligand-receptor pair signals primarily by coupling to $G\alpha_s$ and inducing an increase in cAMP intracellularly that then activates the protein kinase A (PKA) pathway to regulate bone formation and calcium homeostasis[51]. PTH activation of the cAMP-PKA signaling pathway has also been associated with downstream activation of the Wnt/β-catenin signaling pathway that increases bone formation by promoting osteoblast maturation[52]. Interestingly, previous studies from our lab discovered that deletion of Rxfp2 in male mice gubernaculum drastically decreases expression of β-catenin, indicating that it could also be an important target of RXFP2 signaling in bones[53].

We selected compound 6641 as the lead RXFP2 agonist for in vivo testing. Low cytotoxicity suggested the safe use of compound 6641 as a pharmacological agent (Fig. 4b, c, Supplementary Fig. 14), while pharmacokinetic studies showed bone exposure and bioavailability after oral gavage (Supplementary Figs. 9, 12). Previously it was established that C57BL/6 J female mice start losing trabecular bone mass naturally after they reach 8 weeks of age, which provides a model to study the anabolic effects of compound on bone metabolism[54]. The use of females provides a test model with low if any basal INSL3 exposure. Micro-CT analysis of the lumbar spine after an 8-week treatment administered orally 3 times a week demonstrated the ability of compound 6641 to improve trabecular bone parameters, especially Tb.N and Tb.Th (Fig. 5a), but no changes were found in femur trabecular bone (Supplementary Fig. 13a) or the biomechanical properties of either bone (Supplementary Fig. 15). Similar results were observed in the trabecular bone of the lumbar spine of INSL3 transgenic female mice, although with a greater increase in BV/TV and striking difference in Conn.D and Mat.D (Supplementary Fig. 11a), which are comparable with the extent of trabecular changes observed in the lumbar spine after treatment of C57BL/6 J female mice with current PTH analogs prescribed for osteoporosis[55]. In the transgenic model, constant INSL3 exposure since birth could account for these differences, which suggest that increasing the frequency and dosage of 6641 administration could significantly improve the overall trabecular bone mass. Quantitative RT-PCR analysis of tibias from INSL3 transgenic and 6641-treated mice also revealed that there may be differences in the molecular mechanisms activated to trigger the increase in bone formation. As previously reported, INSL3 induces an increase in Alp in primary human osteoblasts[20] (Supplementary Fig. 11b) while 6641 induces an increase in Osteocalcin expression (Fig. 5b).

It should be noted that the proof-of-principle studies with animal models presented here have some limitations as described above. The optimization of compound dosage, vehicle, delivery route, target exposure, as well as further testing in various models of cryptorchidism and diseases associated with bone loss are required to establish the pharmacological utility of identified RXFP2 agonists.

In conclusion, we have successfully developed the orally bioavailable small molecule agonists of the RXFP2 receptor with lead compound 6641. This compound is an allosteric agonist of RXFP2, highly potent, specific, and demonstrates strong mineralization activity in cell-based assays. The efficacy study in vivo showed effects in gubernacular development and in bone maintenance. Additional studies will be needed to fully evaluate the therapeutic potential of this compound in human diseases.

## Methods

**cAMP assays in transfected cell lines.** The HTRF Gs dynamic cAMP assay kit (CisBio, Bedford, MA) was used to measure intracellular cAMP accumulation in HEK293T cells (ATCC, Manassas, VA)[56]. HEK293T cells do not express endogenous RXFP2 or RXFP1 and are thus ideal for specificity testing using transfected receptor constructs. The cell stock was verified mycoplasma free and authenticated by ATCC. Transient transfection was accomplished by treating cells in a 6-well plate with a mixture of 2 μg plasmid and 6 μL Lipofectamine 2000 (Invitrogen, Waltham, MA) per manufacturer's protocol and incubating overnight. For cAMP measurement in 96-well format, HEK293T cells stably transfected with human RXFP2 (HEK-RXFP2), human RXFP1 (HEK-RXFP1), mouse Rxfp1 (HEK-Rxfp1) or transiently transfected with human RXFP1-RXFP2 chimeric receptors were seeded between 7500 and 30,000 cells per well, depending on cell line responsiveness, in serum-free DMEM medium and incubated overnight at 37 °C, 5% $CO_2$. After 18 h, cells were treated with a titration of compound, 2 μM forskolin, or DMSO vehicle. Control wells were also treated with a titration of INSL3 (Phoenix Pharmaceuticals, Burlingame, CA) or human recombinant relaxin 2 (Peprotech, Cranbury, NJ). For the RXFP2 antagonist experiment, cells were co-treated with 0.28 μM compound or 30 nM INSL3 and a titration of the INSL3 B dimer antagonist[38,39]. For the INSL3 allosteric modulation experiment, cells were co-treated with 0.025, 0.0025, or 0.00025 μM of compound and a titration of INSL3. Cells were incubated for 1 h at 37 °C, 5% $CO_2$, and then kit cAMP-d2 and anti-cAMP antibody were added per manufacturer protocol. Plates were incubated for 1 h at room temperature and the signal was read on a CLARIOstar plate reader (BMG Labtech, Ortenberg, Germany) at 665 and 620 nm emission wavelengths.

Experiments to analyze the cAMP response in RXFP3, RXFP4, and ADORA1 transiently transfected HEK293T cells was performed in the National Institute of Mental Health's Psychoactive Drug Screening Program, Contract #HHSN-271-2018-00023-C (NIMH PDSP). As a positive control, Relaxin 3 peptide was used for RXFP3 and RXFP4 and 5′-(N-Ethylcarboxamido)adenosine (NECA) for ADORA1 testing. cAMP production was induced by treatment with 0.1 μM isoproterenol and the effect of Relaxin 3, NECA, or 6641 was measured using the GloSensor™ cAMP Assay (Promega, Madison, WI).

The activation of FSHR, LHCGR, TSHR, and PTHR1 receptors by corresponding peptide hormone or compound 6641 was performed using cAMP Hunter cell lines and HitHunter cAMP XS + assay in Eurofins DiscoverX (Fremont, CA).

**CRE-Luciferase assay in HEK293T cell lines.** HEK293T-CRE-Luc cells express luciferase under the control of the cAMP response element (CRE), which allows for indirect measurement of intracellular cAMP accumulation[57]. HEK293T-CRE-Luc cells were stably transfected with human RXFP2 (HEK-CRE-Luc-RXFP2) or transiently transfected as above with mouse Rxfp2 (HEK-CRE-Luc-Rxfp2). HEK-CRE-Luc-RXFP2 and HEK-CRE-Luc-Rxfp2 cells were seeded at 15,000 and 30,000 cells per well, respectively, in 96-well plates using serum-free DMEM medium and incubated overnight at 37 °C, 5% $CO_2$. After 18 h, cells were treated with a titration of compound, a titration of INSL3, 5 μM forskolin or DMSO vehicle. After treatment, cells were incubated for 3 h at 37 °C, 5% $CO_2$. Plates were then allowed to equilibrate for 30 min at room temperature and an equal volume of substrate (Amplite Luciferase Reporter Gene Assay Kit, AAT Bioquest, Sunnyvale, CA) was added to each well. Plates were incubated for 40 min at room temperature protected from light and the luciferase output was read on a CLARIOstar plate reader.

**PRESTO-Tango assay in HTLA cells transfected with GPCRs.** The PRESTO-Tango GPCRome assay was used to perform a HTS of related GPCRs to study specificity and selectivity of the RXFP2 agonists by measuring β-arrestin translocation activity. The analysis was completed in the National Institute of Mental Health's Psychoactive Drug Screening Program at the University of North Carolina at Chapel Hill. The assay used transiently transfected HTLA cells that stably

express the β-arrestin2-TEV protease fusion protein and a luciferase reporter gene under the control of the Tta transcription factor.

**Receptor expression flow cytometry**. All receptor constructs are expressed within the pcDNA3.1/Zeo(+) AmpR mammalian expression vector and contain a FLAG epitope at the N-terminus[40]. HEK293T cells were harvested in 5 mM EDTA 24 h after transfection and fixed for 10 min in 3.7% formaldehyde/PBS. Cells were then washed twice with surface detection buffer (2% FBS, TBS, 1 mM CaCl₂) or permeabilization buffer (surface detection buffer + 0.2% Tween-20) and incubated with 5 µg/mL anti-FLAG M1 Ab (F3040, Sigma-Aldrich, Burlington, MA) in 100 µl surface detection buffer for 1 h at 4 °C. Cells were washed again with surface detection or permeabilization buffer and incubated with 10 µg/mL Alexa Fluor 488 goat anti-mouse IgG (A11001, Invitrogen, Waltham, MA) in 100 µl surface detection buffer for 20 min at 4 °C protected from light. Cells were washed one last time and resuspended in surface detection buffer for analysis on an Accuri C6 flow cytometer (BD Biosciences, Franklin Lakes, NY). Cell debris were gated out based on FSC/SSC position, and the FLAG-positive population was determined relative to cells transfected with empty pcDNA3.1/Zeo(+) vector.

**Mineralization assay**. Primary human calvarial osteoblast (HCO) cells (ScienCell, Carlsbad, CA) were seeded at 7000 cells per well in 96-well plates using growth medium (DMEM-F12, 10% FBS, 1X Pen/Strep) and incubated overnight at 37 °C, 5% CO₂. After 24 h, the growth medium was replaced with mineralization medium (growth medium + 10 mM β-glycerophosphate, 50 µg/ml ascorbic acid, 10 nM dexamethasone), and wells were treated with various concentration of compound or DMSO vehicle. Medium and treatments were refreshed every 2–3 days. Mineralization was determined at day 14 by measuring hydroxyapatite deposits in the cell matrix using the fluorescence-based OsteoImage Mineralization Assay (Lonza, Basel, Switzerland), as per manufacturer instructions. Mineralization was quantified using a CLARIOstar plate reader with 492 nm excitation and 520 nm emission wavelengths.

**Cytotoxicity assay**. The ATP CellTiter-Glo Luminescent Cell Viability Assay (Promega, Madison, WI) was used to measure cytotoxicity induced by the compounds in HCO cells and human aortic vascular smooth muscle cells (HAVSMC) (ATCC, Manassas, VA). HCO and HAVSMC cells were seeded in 96-well plates at 3000 and 7000 cells per well respectively. Cells were incubated overnight, treated with compound or DMSO vehicle and incubated for 24 h at 37 °C, 5% CO₂. After incubation, plates were equilibrated for 30 min at room temperature and an equal volume of CellTiter-Glo Reagent was added. Plates were shaken for 2 min, incubated at room temperature for 10 min and the luminescent signal was read on a CLARIOstar plate reader.

**Pharmacokinetics studies**. Pharmacokinetic studies were performed by Pharmaron (Beijing, China) to determine the therapeutic range of compound 6641 in 2-month-old C57BL/6 J female mice. The concentration of compound in plasma, liver, and long bones was detected using liquid chromatography tandem mass spectrometry (LC-MS/MS). Samples were collected after a single intraperitoneal (IP) administration at 30 mg/kg, a single intravenous (IV) administration at 3 mg/kg, a single oral (PO) administration at 10 mg/kg, and a daily PO administration for 3 consecutive days at 10 mg/kg. The compound was formulated in 75% PEG300 + 25% of 40% aq. HPBCD for IV and PO administration and in 60% Phosal + 40% PEG300 for IP administration. All tissue samples were homogenized in water at a tissue weight to water volume ratio 1:3.

**Animal studies**. All animal studies were approved under protocols 19-009 and 20-027 by the FIU Institutional Animal Care and Use Committee and performed following the NIH Guide for the Care and Use of Laboratory Animals. All mice were maintained in the FIU animal facilities in a controlled temperature and humidity environment with 12-h light/12-h dark cycles and ad libitum access to food and water.

To study the ability of the RXFP2 agonists to induce gubernaculum development in female embryos, pregnancy was timed by vaginal plug determination (E0.5). Four-month-old WT C57BL/6 J pregnant females (The Jackson Laboratory, Bar Harbor, ME) were given 6 consecutive IP injections with 30 mg/kg of compound 6641, 4337, and 4340 or vehicle (60% Phosal – 40% PEG300) from embryonic day E12.5 to E17.5. At E18.5, the pregnant females were euthanized by isoflurane inhalation overdose and the embryos were extracted for histological analysis.

For the INSL3 study, an INSL3 transgenic mouse model was used[42]. The mouse *Insl3* gene was fused to a rat insulin II promoter to specifically induce *Insl3* overexpression in mouse pancreatic β-cells. INSL3 transgenic female hemizygous mice were mated with WT FVB/N males to obtain hemizygotes and WT littermate offspring. WT and INSL3 transgenic females, as well as WT male littermates were fasted overnight and euthanized by isoflurane inhalation overdose at 22 weeks of age.

For the RXFP2 agonist study, WT C57BL/6 J female mice were used. 8-week-old female mice were randomly allocated into 2 treatment groups (vehicle or compound). Mice were treated 3 times per week with RXFP2 agonist (compound

6641) or vehicle formulation (75% PEG 300 + 25% of 40% aq. HPBCD) for a total of 8 weeks by oral gavage at a concentration of 10 mg/kg using flexible PTFE sterile plastic feeding needles and euthanized by isoflurane inhalation overdose at 16 weeks of age. All mice were closely monitored, and body weight recorded throughout the study. Kidneys and liver were weighed at necropsy and processed for histology evaluation to assess any potential signs of toxicity. To study the differences in *Rxfp2* gene expression between femur and spine, untreated 8-week-old WT C57BL/6 J female mice were used.

**Genotyping**. All embryos were genotyped for sex determination by PCR using DNA extracted from the tail. The primers mouse Sry forward (5'-GGTGTGGTCCCGTGGTG-3') and mouse Sry reverse (5'-TTTGGGTATTTCTCTCTGTGTAGG-3') were designed to amplify a 174 bp region of the Y chromosomal *Sry* gene. Mice were genotyped for the *Insl3* transgene by PCR using DNA extracted from a small piece of the mouse ear. The primers rat Ins2 forward (5'-CTCTGAGCTCTGAAGCAAGCA-3') and rat Ins2 reverse (5'-CCCAG GAGGCTGCAGTTGTT-3') were designed to amplify a 300 bp region of the rat insulin II promoter of *Insl3* transgene[42].

**Serum INSL3 activity determination**. Whole blood was collected by retro-orbital bleeding from all WT and INSL3 transgenic mice and allowed to clot on ice. Serum was collected by centrifugation at 4 °C for 5 min at 2000 rcf and stored at −80 °C until used. To determine the activity of serum INSL3 in the transgenic mice compared to the WT mice, serum was diluted 20-fold in serum-free DMEM and activity on HEK-RXFP2 cells was measured by induction of cAMP using the HTRF Gs dynamic cAMP assay kit (CisBio, Bedford, MA), as described above.

**Micro-CT analysis**. 10% formalin fixed lumbar spine from WT and INSL3 transgenic mice, as well as non-fixed frozen lumbar spine and femur from 6641- and vehicle-treated mice were used for micro-CT analysis. Frozen bones were allowed to thaw for at least 2 h at room temperature before scanning. All micro-CT scan analysis and 3D reconstructions were performed on a Scanco 40 instrument (Scanco Medical, Bassersdorf, Switzerland) using a slice resolution of 12 µm isotropic voxel size. The complete distal femurs from the midshaft portion down and L3–L4 vertebras were scanned with an effective energy of 55 kVp, X-ray tube current of 114 mA, and 200 ms integration time. For the quantification of trabecular bone of the distal femur metaphysis, the region of interest consisted of 151 transverse slices extending 1.8 mm above the distal growth plate. The region of interest for the trabecular bone quantification comprised the entire vertebral body, including the maximum number of slices between both growth plates. All trabecular quantifications were performed applying a grayscale threshold (lower threshold 220, upper threshold 1000) and Gaussian noise filter (sigma 0.8, support 1). Standard nomenclature guidelines were followed to report all micro-CT measurements and the following parameters were calculated: bone volume per tissue volume (BV/TV, %), trabecular thickness (Tb.Th, mm), trabecular number (Tb.N, mm-1), trabecular separation (Tb.Sp, mm), connectivity density (Conn.D (1/mm³) and material density (Mat.D, VOI-mg HA/ccm).

**Biomechanical analysis**. After micro-CT scanning, these same lumbar spine and femur from 6641- and vehicle-treated mice were utilized for biomechanical testing. The femurs were tested in a 3-point bending test and the L3 vertebrae in a compression test using an ElectroForce 5500 Test Instrument (TA Instruments, New Castle, DE) with a ramp rate of 0.05 mm/sec and support span of 8.1 mm, running WinTest software version 8.2. Time, force, and displacement were collected for use in the analysis of mechanical and material properties using custom MATLAB codes (MathWorks, Natick, MA). The Moment of Inertia (MoI) was calculated based on the micro-CT measurements. The elastic modulus (MPa), yield stress (MPa), and ultimate stress (MPa) were calculated for each sample.

**Histological analysis**. Whole embryos, kidneys and livers were fixed in 10% formaldehyde for 24 h, embedded in paraffin wax and 5–7 µm sagittal serial sections were cut using the RM2255 automated rotary microtome (Leica, Wetzlar, Germany). Sections were stained with Harris hematoxylin and eosin and representative images were taken using an AXIO Scope.A1 microscope connected to an Axiocam MRc5 camera (Zeiss, Jena, Germany).

**RNA extraction, cDNA generation and quantitative RT-PCR procedure**. Tibias and lumbar spine were removed from muscle tissue, bone marrow flushed with cold PBS and clean tibias frozen at −80 °C until use. Frozen bones were then processed in TRIzol reagent (Ambion, Austin, TX) for 1 min at 15,000 rpm using a tissue homogenizer. RNA extraction was performed using the Direct-zol RNA Miniprep Plus kit (Zymo Research, Irvine, CA) as instructed by the manufacturer. RNA concentrations and purities were measured using a NanoVue spectrophotometer (GE Healthcare, Chicago, IL). cDNA was generated from 1 µg of RNA using the Verso cDNA synthesis kit (Thermo Fisher Scientific, Waltham, MA) and quantitative RT-PCR was performed using the GoTaq qPCR master mix (Promega, Madison, WI) following manufacturer instructions. Relative expression values of the specific genes of interest were calculated by the $2^{-\Delta\Delta Ct}$ method using *Rpl13a*

gene as normalizer. The sequences of the primers used were as follows: mouse *Runx2* forward 5'-TCAGCAAAGCTTCTTTTGGGAT-3' and mouse *Runx2* reverse 5'-GGGCTCACGTCGCTCAT-3'; mouse Osteocalcin (*Bglap*) forward 5'-ACCCTGGCTGCGCTCTGTCTCT-3' and mouse Osteocalcin (*Bglap*) reverse 5'-GATGCGTTTGTAGGCGGTCTTCA-3'; mouse *Alp* forward 5'-TCCACCAG-CAAGAA GAAGCC-3' and mouse *Alp* reverse 5'-AAACCCAGACACAAGCAT TCCC-3'; mouse *Col1a1* forward 5'-CCTCAGGGTATTGCTGGACAAC-3' and mouse *Col1a1* reverse 5'-ACCACTTGATCCAGAAGGACCTT-3'; mouse *Rxfp2* forward 5'-TGTGCTCAGATGCCTCAAC-3' and mouse *Rxfp2* reverse 5'-AGGAAGGTGGAGTTCGTTATG-3'; mouse *Rpl13a* forward 5'-CCCTGCTGCTC TCAAGGTT-3' and mouse *Rpl13a* reverse 5'-GGTACTTCCACCCGACCTC-3'.

**Statistics and reproducibility**. Data were analyzed using GraphPad Prism 9 (GraphPad Software, San Diego, CA). One-way ANOVA was used to compare three or more groups followed by a Dunnett's multiple comparison test. Unpaired two-tailed Student's *t*-test was used to determine the statistical difference between the two groups. Values are presented as mean ± SEM of independent experiments. The number of samples and repeats is indicated in figure legends. A value of $p < 0.05$ was considered statistically significant.

**Reporting summary**. Further information on research design is available in the Nature Research Reporting Summary linked to this article.

## Data availability
All data presented in the figures and supplemental material are available in Supplementary Data 2 file.

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

## Acknowledgements
The authors would like to thank Dr. R.A.D. Bathgate and Dr. M.A. Hossain at The Florey Institute of Neuroscience and Mental Health, The University of Melbourne, Australia, for the human and mouse RXFP1, RXFP2 and chimeric constructs, and for the INSL3 B-chain dimer antagonist. PRESTO-Tango and cAMP functional data were generously provided by the National Institute of Mental Health's Psychoactive Drug Screening Program, Contract # HHSN-271-2018-00023-C (NIMH PDSP). The NIMH PDSP is Directed by Bryan L. Roth MD, PhD at the University of North Carolina at Chapel Hill and Project Officer Jamie Driscoll at NIMH, Bethesda MD, USA. We also thank Dr. Xi-Ping Huang (UNC) for his support. This research was supported by the National Institute of Arthritis and Musculoskeletal and Skin Diseases (NIAMS) at the NIH (R01AR070093 to A.I.A.). Support for this research was provided also by the National Center for Advancing Translational Sciences Intramural Research Program. The bone core activities at UAMS were supported by the NIH grant P20 GM125503 from NIGMS. M.E.-L. was supported by Doctoral Evidence Acquisition Fellowship (DEA) and Dissertation Year Fellowship (DYF) from the University Graduate School at Florida International University.

## Author contributions
A.I.A., J.J.M., and I.U.A. designed, guided, and supervised the study. M.F. and E.B. conducted the primary HTRF cAMP HTS of the small molecule library. K.J.W., E.R.G. designed, synthesized, and characterized compounds. M.E.-L., C.M., N.T.S., X.H., M.J.H., and W.Y. conducted the confirmatory HTS and analyzed the data. A.W. and X.X. analyzed the pharmacokinetics studies. M.E.-L. conducted the in vivo experiments and analyzed data. E.M.K. conducted the histological analysis of mouse gubernaculum. M.E.-L., R.M., and J.T.S. conducted the bone analysis. All authors discussed the results, assisted in manuscript preparation, and provided final approval of the submitted work.

## Competing interests
M.E.-L., K.J.W., C.M., M.J.H., N.T.S., X.H., E.B., W.Y., M.F., I.U.A., J.J.M., A.I.A. are the inventors in patent application No. 63/308,768, assigned to NIH, DHHS, and FIU, entitled "Small Molecule Agonists of the Insulin-like (INSL3) Peptide Receptor RXFP2 and Methods of Use Thereof", relating to the discovery and the use of small molecule agonists of RXFP2. The remaining authors declare no competing interests.
