## [Peer Review File · Communications Biology]

Reviewers' comments:

Reviewer #1 (Remarks to the Author):

This paper describes the discovery and properties of new allosteric agonists at the Relaxin Family Peptide 2 receptor (RXFP2). The paper describes the in-vitro pharmacology of their compounds, then explore their in-vivo properties. Significantly they show that their prototype ligand can increase vertebral (but not femur)bone mass and they also provide a mechanism for this in terms of effects on gene expression. Thus the paper provides a complete story from pharmacology to physiology, with potential clinical implications. This should make the paper of considerable general interest. The experiments seem to have been done carefully and I have no issues with their interpretation. The statistical analysis seems valid and the methods look to have sufficient detail.

I do have some comments on the paper

1) The authors demonstrate that 6641 is an allosteric agonist of RXFP2. However, they do not investigate whether it can also act as an allosteric modulator of INSL3. This is important information that I suspect will be important for the in-vivo use of 6641, either clinically or as an experimental tool. I would urge the authors to address this.

2) The authors have shown 6641 activates Gs, but do they have data on any other G-protein (particularly Go)? It would be useful at least to comment on this.

3) Line 110; as the authors comment, the supplementary data shows 6641, 4337 and 4340 all activate the murine receptor as well as the human receptor. However, in the murine assay, 4337 and 4340 appear as partial agonists. This is worth noting; it is interesting but could easily get lost as it is only apparent if the supplementary data is viewed by readers.

Reviewer #2 (Remarks to the Author):

The manuscript by Esteban-Lopez et al is a broad reaching and interesting description of the discovery of a small molecule agonist of the RXFP2 receptor. After optimization a small number of molecules were tested for sub-type and species selectivity, broader GPCR selectivity and shown to be allosteric binders. The molecules were shown to induce osteoblast mineralization in vitro and bone formation and gender-specific tissue differentiation in vivo. It was reassuring to see the authors perform PK characterization to back up the in vivo work.

There are some minor points I feel need to be addressed:

1 Please mention it is the human RXFP2 receptor in the main text (already mentioned in experimental)

2 Cross screening against other GPCRs was done using the PRESTO-Tango system but this is only a beta-arrestin readout, so G protein activation could easily be missed. In addition it is also stated that the compounds don't activate arrestin signalling on RXFP2, so to use arrestin signalling as an indicator of selectivity is not that valid. Further I think the way the assay is configured would not detect antagonists at any of the receptors. This section needs to be addressed before publication.

3 The authors demonstrate the molecules are allosteric by use of chimeric receptors combined with the fact that the natural agonist binds to the extra-cellular domain. This is fine, but it probably means the molecules bind in what is the orthosteric site in other family A GPCRs. It would also be interesting as to whether the molecules are also positive modulates (i.e. can enhance signalling by the peptide). This wasn't tested; I suggest this possibility is mentioned in the manuscript and can be followed up later.

Overall it is a strong piece of work and merits publication.

Reviewer #3 (Remarks to the Author):

The authors discovered allosteric agonists of GPCR RXFP2 and did systematic study to investigate

their properties. I only have a minor suggestion about the RXFP family:

The value of sequence identity between RXFP1 and RXFP2 should be listed (line 115). Also, there are two other members in RXFP family, RXPF3 and RXPF4. Similarity among all these members could be mentioned.

Reply to reviewers' comments

Reviewer #1(Remarks to the Author): This paper describes the discovery and properties of new allosteric agonists at the Relaxin Family Peptide 2 receptor (RXFP2). The paper describes the in-vitro pharmacology of their compounds, then explore their in-vivo properties. Significantly they show that their prototype ligand can increase vertebral (but not femur)bone mass and they also provide a mechanism for this in terms of effects on gene expression. Thus the paper provides a complete story from pharmacology to physiology, with potential clinical implications. This should make the paper of considerable general interest. The experiments seem to have been done carefully and I have no issues with their interpretation. The statistical analysis seems valid and the methods look to have sufficient detail. I do have some comments on the paper 1) The authors demonstrate that 6641 is an allosteric agonist of RXFP2. However, they do not investigate whether it can also act as an allosteric modulator of INSL3. This is important information that I suspect will be important for the in-vivo use of 6641, either clinically or as an experimental tool. I would urge the authors to address this.	Thank you for your positive comments! 1) We have performed a series of experiments to investigate the possible allosteric modulation of the lead compound with native ligand INSL3. The results are shown in new Supplementary Fig.8 where we analyzed the cAMP response after treatment of HEK293T cells transfected with RXFP2 with variable concentrations of both ligands. No positive or negative modulation of the cAMP response was detected in co-treatment with two ligands.
---	---

2) The authors have shown 6641 activates Gs, but do they have data on any other G-protein (particularly Go)? It would be useful at least to comment on this.

3) Line 110; as the authors comment, the supplementary data shows 6641, 4337 and 4340 all activate the murine receptor as well as the human receptor. However, in the murine assay, 4337 and 4340 appear as partial agonists. This is worth noting; it is interesting but could easily get lost as it is only apparent if the supplementary data is viewed by readers.

Reviewer #2 (Remarks to the Author):

The manuscript by Esteban-Lopez et al is a broad reaching and interesting description of the discovery of a small molecule agonist of the RXFP2 receptor. After optimization a small number of molecules were tested for sub-type and species selectivity, broader GPCR selectivity and shown to be allosteric binders. The molecules were shown to induce osteoblast mineralization in vitro and bone formation and gender-specific tissue differentiation in vivo. It was reassuring to see the authors perform PK characterization to back up the in vivo work.

There are some minor points I feel need to be addressed:

1. Please mention it is the human RXFP2 receptor in the main text (already mentioned in experimental)

2) It is a valid point and as suggested we commented in Discussion on the possibility of biased G protein signaling by small molecule vs native hormone.

3) It is correct, the lead compound indeed behaves as a full agonist whereas the other two tested compounds are partial agonists vs mouse receptor. This is clarified in the text.

1. Corrected in the text of the manuscript. Please note however that as our experiments in vitro and in vivo showed that the lead compound 6641 is also a full agonist of mouse RXFP2 receptor.

2. Cross screening against other GPCRs was done using the PRESTO-Tango system but this is only a beta-arrestin readout, so G protein activation could easily be missed. In addition it is also stated that the compounds don't activate arrestin signalling on RXFP2, so to use arrestin signalling as an indicator of selectivity is not that valid. Further I think the way the assay is configured would not detect antagonists at any of the receptors. This section needs to be addressed before publication.

3. The authors demonstrate the molecules are allosteric by use of chimeric receptors combined with the fact that the natural agonist binds to the extra-cellular domain. This is fine, but it probably means the molecules bind in what is the orthosteric site in other family A GPCRs. It would also be interesting as to whether the molecules are also positive modulates (i.e. can enhance signalling by the peptide). This wasn't tested; I suggest this possibility is

2. We presented new data showing that in addition to RXFP1, two other members of RXFP family, RXFP3 and RXFP4 receptors, were not activated by the lead compound. Please note that an activation of RXFP3/4 receptors by cognate ligand, relaxin 3 peptide, is mediated through G α i and induces an inhibition of adenylate cyclase and, therefore, a decrease in the cAMP concentration. We also tested the three LRR-containing glycoprotein hormone receptors most similar to RXFP2, FSHR, LHCGR, and TSHR, as well as PTHR1 receptor, a member of the Class B GPCRs. All 4 receptors signal through G α s and cause increased cAMP production. None of these receptors were activated by our lead compound. As we showed before, ADORA1 receptor signaling through G α i is not activated by compound either. The GPCRome PRESTO-Tango system was used to highlight an absence of off-target binding to other GPCRs. Indeed, compound antagonist function cannot be detected in this assay. We mentioned this important issue in Discussion. See Supplementary Figs.3 and 4.

3. The analysis of the compound binding sites in RXFP2 is currently ongoing and will be finalized shortly. The preliminary data however indicate that the binding pocket for 6641 is different from the one utilized by ML290 small molecule agonist in the RXFP1 receptor. As to analyzing if 6641 modulates INSL3/RXFP2 signaling, we did perform additional experiments to detect such interaction. Please see above the reply to the similar

mentioned in the manuscript and can be followed up later. Overall it is a strong piece of work and merits publication. Reviewer #3 (Remarks to the Author): The authors discovered allosteric agonists of GPCR RXFP2 and did systematic study to investigate their properties. I only have a minor suggestion about the RXFP family: 1. The value of sequence identity between RXFP1 and RXFP2 should be listed (line 115). Also, there are two other members in RXFP family, RXPF3 and RXPF4. Similarity among all these members could be mentioned.	comment by Reviewer#1. See Supplementary Fig. 8. Thank you! Thank you! 1. We included sequence identity values between RXFP2 and RXFP1 receptors (56%) as well as with RXFP3 and RXFP4 (about 10%). We also performed analysis of cAMP activation by compound 6641 in three related LRR-containing glycoprotein hormone receptors, FSHR, LHCGR, and TSHR which show between 23-31% amino acid identity with RXFP2. None were activated by 6641. We also tested and included in the manuscript the analysis of a more distant representative of Class B GPCRs, PTHR1. See new Supplementary Figs 3 and 4.
--	--

REVIEWERS' COMMENTS:

Reviewer #1 (Remarks to the Author):

I am satisfied with the responses of the authors to my comments

Reviewer #2 (Remarks to the Author):

The manuscript by Esteban-Lopez et al is a broad reaching and interesting description of the discovery of a small molecule agonist of the RXFP2 receptor. After optimization a small number of molecules were tested for sub-type and species selectivity, broader GPCR selectivity and shown to be allosteric binders. The molecules were shown to induce osteoblast mineralization in vitro and bone formation and gender-specific tissue differentiation in vivo. It was reassuring to see the authors perform PK characterization to back up the in vivo work.

I had previously raised some points concerning the manuscript. These have all been addressed satisfactorily and this represents a very interesting piece of work.

Reviewer #3 (Remarks to the Author):

It is great that the authors added the data I suggested.